# The Lack of Effect of Botulinum Toxin-A on Upper Limb Activity in Chronic Stroke: A Short Report from the InTENSE Trial

**DOI:** 10.3390/toxins16120510

**Published:** 2024-11-26

**Authors:** Ian D. Cameron, Louise Ada, Maria Crotty, Mithu Palit, Lydia Huang, John Olver, Steven G. Faux, Senen Gonzales, Brian Anthonisz, Malcolm Bowman, Yuriko Watanabe, Yan Chow, Rachel Milte, Julie Ratcliffe, Coralie K. English, Natasha A. Lannin

**Affiliations:** 1John Walsh Centre for Rehabilitation Research, Northern Sydney Local Health District, St Leonards, NSW 2065, Australia; ian.cameron@sydney.edu.au; 2Faculty of Medicine and Health, University of Sydney, Camperdown, NSW 2050, Australia; 3College of Medicine and Public Health, Flinders University, Bedford Park, SA 5042, Australia; 4Flinders Medical Centre, Bedford Park, SA 5042, Australia; 5Alfred Health, Melbourne, VIC 3004, Australia; 6Epworth Monash, Rehabilitation Research Unit, Richmond, VIC 3121, Australia; 7Sacred Heart Rehabilitation, St Vincent’s Hospital Sydney, Darlinghurst, NSW 2010, Australia; 8St Vincent’s Clinical School, University of New South Wales, Sydney, NSW 2033, Australia; 9Austin Hospital, Heidelberg, VIC 3084, Australia; 10St Joseph’s Hospital, St Vincent’s HealthCare, East Sydney, NSW 2010, Australia; 11Caring Futures Institute, Flinders University, Bedford Park, SA 5042, Australia; 12Heart and Stroke Program, School of Health Sciences, University of Newcastle, Callaghan, NSW 2308, Australia; 13Hunter Medical Research Institute, New Lambton Heights, NSW 2305, Australia; 14Department of Neurosciences, Central Clinical School, Monash University, Clayton, VIC 3800, Australia

**Keywords:** botulinum toxin type-A, movement pain, quality of life, spasticity, wrist, rehabilitation

## Abstract

We examined the effect of botulinum toxin-A on upper limb impairments and activity limitations in chronic stroke. This study is a secondary analysis of control group data from a national, multicenter, Phase III randomized trial with a masked outcome assessment. We studied 71 stroke survivors who received a botulinum toxin-A injection in any muscle(s) that crosses the wrist due to significant spasticity after a stroke greater than 3 months previously. We measured upper limb activity, spasticity, range of motion, grip strength, pain and other outcomes at injection and three months later. The median difference between injection and 3 months later was 0.0 blocks/s (interquartile range (IQR) 0.0) on the Box and Block Test, 0/4 (IQR 1) on the Tardieu Scale, 4 degrees (IQR 26) of wrist extension, 0.0 kg (IQR 2) of grip strength, 0.0 (IQR 1.5) on the 10 cm visual analogue scale for pain, 0/100 (IQR 21) on the 10 cm visual analogue scale for overall health, 0/3 (IQR 0) for self-care and −2 (IQR 8) for burden of care. In chronic stroke survivors who have little activity in their upper limb, botulinum toxin-A is not effective in improving any measured outcomes and does not appear to be clinically justified in this population with severe activity limitations.

## 1. Introduction

There is a continuing debate about the effectiveness of botulinum toxin in stroke, particularly with reference to improving activity, meaning the execution of a task or action by an individual. Australian guidelines suggest that this intervention is unlikely to improve activity or motor function [1], whereas UK guidelines suggest that it is a safe and effective intervention [2]. Our findings from the InTENSE trial showed that multidisciplinary intervention in addition to botulinum toxin in people with chronic stroke and disability of the upper extremity did not improve activity and had a marginal effect on other outcomes [3]. However, it has been reported by others that botulinum toxin was associated with “moderate improvement in upper-extremity activity capacity or performance after stroke” [4]. For this brief report, our aim is to examine data from the control group of the InTENSE trial and determine the effect of botulinum toxin-A alone in muscles that cross the wrist.

## 2. Methods

This study examines data from the control group of the InTENSE trial at the time of injection and three months later. Patients were included if they were adults over 3 months post-stroke, were scheduled to receive a botulinum toxin-A injection to a muscle(s) that crosses the wrist and were not currently receiving upper limb rehabilitation [3,5].

Participants had a standard injection program according to Australian practice recommendations [6]. Structured goal setting with participants and their carers was completed within the spasticity clinic, and rehabilitation physicians selected the muscle(s) for injection before inclusion in the trial. Botulinum toxin-A was supplied to participants through the Australian Pharmaceutical Benefits Scheme [7], and the maximum dose at any time point was 600 units (maximum volume per site = 0.5–1.0 mL). The actual dose was determined by the treating physician. The trial used one neurotoxin, Botox (100 unit vial), which is approved for use for upper-limb spasticity after stroke in Australia. Participants then received a handout (7 stretches and 8 arm and hand exercises), one follow-up telephone call to encourage practice and a log to record practice.

The outcomes were upper limb activity measured using the Box and Block Test [8], spasticity (Tardieu Scale) [9], wrist range of motion (goniometry), grip strength (dynamometry), pain (10 cm visual analogue scale), burden of care (Carer Burden Scale) [10] and quality of life (EuroQol-5D 3L) [11].

Summary statistics for the outcomes at injection and three months later and change scores were determined. The distributions of the data for all measures were skewed and some (e.g., Box and Block test) were highly skewed, so medians (IQR) are presented. Where there was bias between the lower and upper quartiles of the distributions (e.g., pain, spasticity, carer burden and range of movement), bootstrapping using 1000 iterations was used to calculate 95% confidence intervals.

## 3. Results

A total of 71 control participants were recruited from seven spasticity clinics (average throughput of 26 (SD 12) people with stroke per clinic per year) across three states in Australia. The mean age of the participants was 60 years, 70% were male, their stroke occurred a median of 3.3 years previously, 46% had independent ambulation and 52% had past injections of botulinum toxin-A. There was no loss to follow-up at 3 months but there was missing spasticity data for two participants. The most commonly injected muscles were the Flexor Digitorum Superficialis, Flexor Carpi Ulnaris and Flexor Digitorum Profundis, and each participant received a mean of 146 units of botulinum toxin-A.

The median difference between injection and 3 months later was 0.0 blocks/s (interquartile range (IQR) 0.0) on the Box and Block Test, 0/4 (IQR 1) on the Tardieu Scale, 4 degrees (IQR 26) of wrist extension, 0.0 kg (IQR 2) of grip strength, 0.0 (IQR 1.5) on the 10 cm visual analogue scale for pain, 0 (IQR 21) on the 100 mm visual analogue scale for overall health, 0 (IQR 0) for self-care and −2 (IQR 8) for burden of care (Table 1). On further inspection, few participants with no upper limb activity at injection (6 of 71) improved in terms of the Box and Block Test. Nine participants whose performance deteriorated balance this out. We therefore carried out a responder analysis. The outcomes of participants whose Tardieu score improved by ≥1/4 (i.e., 26 of 69 because there was no Tardieu score for 2 participants at 3 months) showed that there were no clinically relevant differences over time in this group (Table 1). Responders had a median improvement of one Tardieu score grade.

## 4. Discussion

The findings show that in chronic stroke survivors with a severely disabled upper limb, botulinum toxin-A was not effective in improving upper limb activity. In addition, any effect on spasticity had worn off and it was only marginally effective in terms of range of movement at 3 months after injection. There was no clinically significant difference for any of the outcomes. Furthermore, the botulinum toxin-A injections conservatively cost AUD 1600 per person, without taking into consideration the cost of running the spasticity clinic.

The InTENSE trial showed that incorporating botulinum toxin with a multidisciplinary intervention is not effective. Taken together, the findings suggest that botulinum toxin injections in the upper extremity in people with chronic stroke with little activity are not effective, either clinically or in terms of cost. They will have no material benefit to participants in their daily lives. However, it is recognized that botulinum toxin is sometimes used in the upper extremity for reasons other than improving activity.

These findings are at variance with spasticity guidelines [1,2] and the possible reasons for this should be considered. These could be that the patients included have chronic impairments after stroke which are severe and therefore have a limited potential to benefit from the injection. The reasons do not include a lack of input from a multidisciplinary team.

This analysis of a subgroup of the InTENSE trial has both strengths and weaknesses. Its main strength is that the participants are representative of those attending spasticity clinics around Australia and internationally [12] in terms of time post-stroke and muscles receiving injection. However, while representative, they had not had a stroke for a long time and had severe disability. There could be a benefit if botulinum toxin-A was administered earlier after stroke, noting that the current participants had not had a stroke for approximately three years [13]. The findings are in line with the current Australian guidelines in which up to four injections are subsidized in the first year of intervention in the absence of severe contracture and subject to response.

## 5. Conclusions

This analysis of the control group of the InTENSE trial suggests that people with severe disability in the chronic phase after stroke do not benefit from botulinum toxin-A with reference to improving activity or other outcomes. Furthermore, we could not identify a group of responders for whom this intervention could be justified. There should be a reconsideration of the use of this intervention in this group. Resources could be reallocated to other types of intervention that are likely to improve activity and participation, such as task-specific practice and training in the use of appropriate aids [1].

## Figures and Tables

**Table 1 toxins-16-00510-t001:** Median (IQR) for all participants, and median difference (IQR) between times for all participants and responders.

Outcome	Times		Difference Between Times (Month 3 Minus Month 0)	
	Month 0	Month 3	All n = 69	Responders n = 26
Activity				
Upper limb activity according to Box and Block Test blocks/s	0 (0.17)	0 (0.17)	0 (0)	0 (0.04)
Impairments				
Spasticity according to Tardieu Scale, 0–4	2 (0)	2 (0)	0 (1)	−1 (1)
Wrist extension, deg	43 (37)	51 (36)	4 (26)	1 (21)
Pain, 0–10 cm	0.3 (3.7)	0 (1.3)	0 (1.5)	−0.1 (1.9)
Grip strength dynamometry, kg	2 (8)	2 (8)	0 (2)	0 (2)
Other				
Quality of life, EuroQual-5D				
Overall health, 0–100	60 (34)	65 (30)	0 (21)	0 (27)
Self-care, 0–3	2 (0)	2 (0)	0 (0)	0 (0)
Burden of care, Care Burden Scale, 0–16	6 (5)	4 (5)	−2 (8)	−2 (5)

## Data Availability

The data presented in this study are available on request from the corresponding author, as approval for raw data release has not been approved by the Ethics Committee.

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
