# Peer review of "The Lack of Effect of Botulinum Toxin-A on Upper Limb Activity in Chronic Stroke: A Short Report from the InTENSE Trial"

_toxins, 2024, doi:10.3390/toxins16120510_

Round 1
Reviewer 1 Report
Comments and Suggestions for Authors
The abbreviation IQR should be stated in full early on.
In line 52, the authors state the maximum dose was 600 units. In line 79, they state the mean dose was 146 units. Does this mean that patients received a very wide range of doses? If so, how were the doses selected for each patient?
The authors have defined a group of “responders” (26 patients) whose Tardieu score improved by greater than or equal to 1 out of 4. But Table 1 shows this group had a median difference between time 0 and 3 months as -1. This should be explained.
The authors have not discussed the possible effect of increasing dose in the patients treated. No mention is made of varying the dose. This should be discussed, even though this is a short report.
Author Response
Please see attachement

Reviewer 2 Report
Comments and Suggestions for Authors
The manuscript is a follow up on InTENSE trial evidencing further results obtained from the control group, consisting in stroke patients that obtained a Botulinum toxin A injection in a muscle crossing the wrist, due to spasticity, and no upper limb rehabilitation.
Similarly to what was published in the InTENSE trial manuscript (doi: 10.1161/STROKEAHA.119.027602) here authors highlighted that patients in the control group showed no significant improvement in the upperlimb function 3 months after Botulinum toxin A injection. As suggested, this might be related to the fact that patients included in the trial present very severe motor dysfunction possibly resulting from a severely damaged corticospinal tract. Therefore, the following questions emerge, which should be considered prior to publication:
- In the absence of a regenerative intervention was it expected some degree of motor function recovery after botulinum toxin A injection? What is the rationale and how does that relate to the mechanims of action of botulinum toxin A (neurotransmission blocker)?
- Clarification of responders vs non-responders is needed in the manuscript text.
- Were any electromyography assessments done throughout the study? Do authors consider that a selection of patients for Botulinum toxin A injection based on muscle activity recordings would be a better approach to identify possible responders?
Author Response
Please see attachement
